# The Feasibility and Effectiveness of a New Practical Multidisciplinary Treatment for Low-Back Pain: A Randomized Controlled Trial

**DOI:** 10.3390/jcm9010115

**Published:** 2019-12-31

**Authors:** Pia-Maria Wippert, David Drießlein, Heidrun Beck, Christian Schneider, Anne-Katrin Puschmann, Winfried Banzer, Marcus Schiltenwolf

**Affiliations:** 1Sociology of Health and Physical Activity, University of Potsdam, 14469 Potsdam, Germany; apuschma@uni-potsdam.de; 2Department of Health Sciences and Technology, Laboratory of Movement Biomechanics, ETH Zürich, CH-8092 Zurich, Switzerland; 3Statistical Consulting Unit StaBLab, Ludwig-Maximilians-Universität München, 80539 Munich, Germany; david.driesslein@gmail.com; 4University Hospital Carl Gustav Carus at Technical University Dresden, 01307 Dresden, Germany; Heidrun.Beck@uniklinikum-dresden.de; 5Orthopädiezentrum Theresie, 80339 München, Germany; 6Department of Sports Medicine, Goethe University Frankfurt, 60323 Frankfurt am Main, Germany; banzer@med.uni-frankfurt.de; 7Pain Management, Centre of Orthopaedics and Trauma Surgery, Conservative Orthopaedics and Pain Management, Heidelberg University Hospital, 69120 Heidelberg, Germany; marcus.schiltenwolf@med.uni-heidelberg.de

**Keywords:** low-back pain, multidisciplinary pain treatment, sensorimotor exercise training, classical conditioning, executive function, MiSpEx Network

## Abstract

Low-back pain is a major health problem exacerbated by the fact that most treatments are not suitable for self-management in everyday life. Particularly, interdisciplinary programs consist of intensive therapy lasting several weeks. Additionally, therapy components are rarely coordinated regarding reinforcing effects, which would improve complaints in persons with higher pain. This study assesses the effectiveness of a self-management program, firstly for persons suffering from higher pain and secondly compared to regular routines. Study objectives were treated in a single-blind multicenter controlled trial. A total of *n* = 439 volunteers (age 18–65 years) were randomly assigned to a twelve-week multidisciplinary sensorimotor training (3-weeks-center- and 9-weeks-homebased) or control group. The primary outcome pain (Chronic-Pain-Grade) as well as mental health were assessed by questionnaires at baseline and follow-up (3/6/12/24 weeks, M2-M5). For statistical analysis, multiple linear regression models were used. N = 291 (age 39.7 ± 12.7 years, female = 61.1%, 77% CPG = 1) completed training (M1/M4/M5), showing a significantly stronger reduction of mental health complaints (anxiety, vital exhaustion) in people with higher than those with lower pain in multidisciplinary treatment. Compared to regular routines, the self-management–multidisciplinary treatment led to a clinically relevant reduction of pain–disability and significant mental health improvements. Low-cost exercise programs may provide enormous relief for therapeutic processes, rehabilitation aftercare, and thus, cost savings for the health system.

## 1. Introduction

Low-back pain (LBP) is a fundamental health burden and a global challenge in an aging and increasing world population [1]. Lifestyle changes such as sedentary occupations or obesity trends will amplify this further. Most studies, conducted in high-income countries, only give recommendations for treatment rather than for prevention. Questions regarding cost-effective population-based medical supply in rural and lower-income areas remain open [2]. In Germany, a country with around 83 million people, the estimated annual costs for treatment of LBP are between 16 and 22 billion Euros [3] and are likely to be higher if work absence is taken into account. As a result, many efforts are made to develop effective therapies for LBP [4].

Therapy guidelines for nonspecific LBP point out that biopsychosocial factors should be taken into account as early as possible [5,6] and that medication, surgeries, and imaging should be handled prudently [7]. They also point to comprehensive self-training and psychological treatment options for those affected. This is especially necessary for people with higher pain levels. Although interdisciplinary programs are more effective in reducing LBP [8,9,10] than unimodal treatments [11,12], their long-term effects remain limited. Furthermore, there is hardly any knowledge on prevention, cost efficiency, and implementation of simple and evidence-based programs that can be carried out at home. There are several explanations for this:

Firstly, most interdisciplinary programs are not suitable for self-management in everyday life. In general, existing interdisciplinary therapies involve multidisciplinary treatment teams and consist of several weeks of intensive therapy, requiring significant time and other resources from both the patient and provider. Although addressing all the underlying components of LBP earlier in the chronification process may yield broader benefits for both patient and the health care systems, interdisciplinary therapies tend to only be available for patients in the advanced stages of LBP. Secondly, the widely accepted biopsychosocial model of chronic pain [13] not only suggests that biological, social, and psychological factors contribute to the development of chronic pain, but also that there are interactions between these dimensions [14]. Therefore, an interdisciplinary intervention that takes into account these interactions is likely to be more effective. Nonetheless, in most existing interdisciplinary therapies, the separate components have not been combined in order to achieve mutually reinforcing effects. Thirdly, as LBP is heterogeneous and people vary in their chronicity, pain intensity, activity level, and functional level, treatments should be tailored to the individual’s capabilities and specific requirements of the patient, which most treatments do not do.

In this paper, the feasibility of a new multidisciplinary treatment for LBP is described that aims to address the mentioned limitations of lacking reinforcing components. It should be evaluated whether (1) persons with higher pain in multidisciplinary intervention improve more than those with lower pain regarding pain and mental health and whether (2) persons with higher pain in the multidisciplinary intervention group improve more than persons in the control group (with regular routines) regarding pain and mental health.

As studies with isolated therapy modules show mixed results, it is expected that this new low-cost, homebased multidisciplinary exercise intervention with mutually reinforcing therapeutic components may be more consistent in its effects on pain and mental health than regular routines, through which especially persons with higher pain grades may benefit more.

## 2. Materials and Methods

### 2.1. Participants

The study took place in orthopedic outpatient clinics across Germany. Patients contacting the clinic were informed about the study after their medical consultation. An open recruitment was chosen to be able to assess preventive and curative effects of the intervention. Eligible patients were between 18 and 65 years. Back pain was defined as a minimum score of 20 on a 100-point Visual Analogue Scale. Exclusion criteria were infections (lasting longer than 7 days), pregnancy, not being able to stand upright independently, not being able to get up from a lying position, inability to fill out questionnaires independently, the presence of illnesses/syndromes that contraindicate physical activity, acute back pain that started 7 days prior to study inclusion, and participation in another study from the same research consortium. In total, *n* = 744 patients signed up for the study after receiving both written and oral information. Sample size calculation was based on a pilot study (unpublished material) in which the largest effect size for the outcome pain was selected (*α* ≤ 0.05; 1-*β* = 0.999, drop out 30%, power analysis by G*Power [15]). For this study, effect size was conservatively completed to *f* = 0.25.

### 2.2. Design and Procedure

Study objectives were analyzed in two arms of a single-blind three-armed randomized controlled trial conducted by the MiSpEx network (design see [16,17,18]). In this network, several thousand people were examined over 8 years in three multicenter studies and 34 partial studies [17]. The objectives of the presented paper were treated in the second multicenter study. In this, participants were randomly allocated (*n*_block_ = 18, basis 1:1; www.randomization.com) to a unimodal (SMT, solely motor control exercise), a multidisciplinary (SMT + BT, motor control exercise and behavioral treatment), and a control group (CG, regular routines in ambulatory settings) shown in Figure 1. Study personnel were blinded, and participants were told not to communicate their group allocation to other participants or study staff. All clinical investigations have been conducted according to the principles expressed in the Declaration of Helsinki. Final ethical approval was provided on 01/25/2012 by the major institutional ethics review board of the University of Potsdam, Germany (number 36/2011). Measurement during this twelve-week intervention program took place at the beginning of the program (M1), after 3 (M2), 9 (M3), 12 (M4), and 24 weeks (M5). In each measurement, questionnaires (for an assessment of psychosocial risk factors and mental health), physical examination, and biomechanical measurements (for an assessment of physical function) were applied (for an overview of the overall study design, see [19]). The study was registered as a clinical trial on 05/16/2013 in the German Clinical Trial Register with the identification number: DRKS00004977. Study conduction was between 06/2013 and 12/2014. All adverse events were reported in the manuscript.

Due to the consideration that multidisciplinary therapy is particularly indicated for people with higher and more complex pain problems, the study objectives were only realized within and for a comparison between the multidisciplinary and the control group.

### 2.3. Intervention

#### 2.3.1. Sensorimotor Training

Rationale of selection: Although effects of exercising are mostly small and evidence about a specific form of exercise superior to another is lacking, it is suggested that motor control exercise (sensorimotor training, SMT) may be important for LBP patients by enhancing core stability, spine control, and muscle performance [20,21,22,23]. Through activation of the deep trunk muscles, improvement of innervation patterns and intra- as well as intermuscular function, SMT targets the restoration of control and coordination of spine muscles, progressing to more functional stability [24]. This adaptation is influenced/moderated by sensory information from both passive and active structures to the central nervous system, from exercise related central neuroplastic effects [25,26,27] and exercise-induced-hypoalgesia (EIH) [28,29,30]. EIH further moderates pain sensation during exercise, anxiety, and fear-avoidance to exercise, which is often an underestimated problem in the success of and compliance to exercise interventions.

The sensorimotor training (SMT) consisted of two parts: a three-week supervised center-based training program, followed by a nine-week individual homebased training (see also [31,32,33]). In both phases of the program, participants trained three times a week. The center-based program was guided by experienced sports therapists and physiotherapists and took place in small groups of up to six participants. The homebased training was audio-guided via DVD. Each training session took approximately 30 minutes and consisted of four different motor control exercises aiming at training core stability and/or surrounding muscles, and upper and/or lower extremities. The exercises were: (1) quadrupedal/all-fours stability; (2) deadlift/rowing; (3) double leg–single leg heel-pad-stance; and (4) side planks. For each exercise, three sets of ten repetitions were performed with a two-minute break between sets. A graded activity approach was used, and each exercise had twelve difficulty levels. The baseline difficulty level for each participant was determined by the trainer at the beginning of the intervention. Before starting the homebased phase, a target level for the homebased phase was determined for each participant. The participant received a personalized training plan with the weekly level increase required to reach the target level. Participants could contact the trainer at any time during the homebased phase. The control group continued their standard medical care.

#### 2.3.2. Multidisciplinary Intervention

In addition to the SMT, the interdisciplinary intervention group received three supplementary treatment modules. The above described SMT training was supplemented with (1) a cognitive distraction task during the SMT (see also [34,35]); (2) a body scan unit directly after the SMT; and (3) psychoeducation. All three modules were provided via DVD and were delivered as an integrated part of the SMT training (cognitive distraction and body scan unit) or as a separate component (psychoeducation) (see also [19]). 

#### 2.3.3. Rationale of Selection 

Cognitive distraction tasks have been proven to be effective in modulating pain sensation and affect [34,36] and may be helpful as a deconditioning tool for pain sensation and fear-avoidance to exercise [37]. Several experimental studies showed that simultaneously offered (executive) memory working tasks during pain stimulation provoke a reduced pain sensation (intensity) and affective reaction (pleasantness) [38,39,40,41]. Some studies indicated that working memory tasks can be combined with general exercise tasks [35,42], but these studies did not discuss the influence of cognitive tasks on motor amplitude, fear, and functionality in pain patients. For this, a simultaneous presentation of cognitive distraction tasks during SMT exercises seems to be promising to influence pain sensation and to ‘unlearn’ both conscious and unconscious associations between movement and pain (e.g., fear-avoidance or cortical and subcortical interactions). The tasks were audio-guided while performing the SMT exercises and comprised the n-back task ([23]; for SMT-exercises 2 and 3) and an adapted version of the California Verbal Learning Task (CVLT; [43]; for SMT-exercise 4). The first of four SMT exercises was provided without any cognitive distraction technique. In the n-back task, the participant was presented with a series of stimuli and was required to indicate whether the present stimulus matched the one from n steps earlier. As with the sensorimotor training, the cognitive distraction followed a graded activity approach with three difficulty levels (level 1: 1-n-back, first three weeks, level 2: 2-n-back, week four until seven; and level 3: 3-n-back, earliest start at week seven). For the adapted version of the CVLT, participants heard a list of 10 words and were required to repeat them in the same order after a 30 second interval. 

Body Scan/ Bodily Awareness—a subtype of distraction in form of bodily sensations—can lead to a reduction in fear-avoidance behavior and an improved perception of sensory information [44], while increasing neuronal activity and concentration of neurotransmitter [45]. Furthermore, body scans—mostly integrated in a mindfulness-based stress reduction program (MBSR [46])—may support the restructuring of the brain volume and biochemical deteriorations that affect pain patients [47]. As it was shown that MBSR is powerful in increasing stress resilience [48], this association may additionally reinforce the intended pathways through the combination of exercise and distraction [49]. Because stress and pain stimuli are processed in the same brain areas, additional effects on central and peripheral sensitization may therefore be expected [14]. One component of the MBSR—the body scan—is a training unit in which participants focus on different parts of bodily awareness which was shortened to 10 minutes [50] and offered directly after the sensorimotor training. The body scan also had several levels, with the more advanced levels requiring the participants to focus their attention on two or more body parts simultaneously.

Psychoeducational activities, such as information about pain processing and principles of chronification as well as suggestions to family members, support changes in behavioral patterns and help to understand the mechanisms of the pain becoming chronic [51,52]. Psychoeducation would allow participants to understand the beneficial role of exercise in pain, thereby further supporting the interruption of the negative link between movement and pain. In addition, participants would learn about the role of personal factors, such as stress, in the chronification of pain and provide them with ways to cope with this. The psychoeducation module was provided via three films of 45 minutes each that participants were asked to watch at home together with their partners (if applicable) in the first weeks of the intervention (center-based phase). The overall objective of the psychoeducation was to change maladaptive pain cognitions such as fear-avoidance beliefs, stimulate the responsible use of medication, reduce inactivity and stress, and stimulate more adaptive interactions with the social environment. Each film had a specific focus: Part I explained the biopsychosocial model of chronic pain and the role of social and personal risk factors in its etiology (education phase), Part II focused on coping strategies and (provided) tasks to practice these coping strategies (exercise phase), and Part III focused on reflection on and the improvement of daily life routines (transfer phase).

### 2.4. Instruments

Pain was assessed by the Chronic Pain Grade questionnaire (CPG [53]) with the subscales: characteristic pain intensity (CPI: 0 = “no pain” to 100 = “the worst pain imaginable”), and subjective disability (DISS: 0 = “no disability” to 100 = “I was incapable of doing anything”) for the last three months. “Mental Health” was operationalized due to a measurement of stress (PSS [54]), vital exhaustion (VE [55]), and anxiety and depression (HADS-D [56]). Pain-related cognitions were measured by the fear-avoidance beliefs questionnaire (FABQ-D, [57]), and pain vigilance and awareness by the German version of the Pain Vigilance Questionnaire (PVAQ, [58]). Lifestyle factors such as socioeconomic status, alcohol and tobacco consumption, medication, and physical activity status were assessed via standardized questions. Social support was measured via the Berliner Social Support Scales (BSSS, [59]).

### 2.5. Statistical Analysis

Study objectives were evaluated via multiple linear regression models (for objective 1: regression model for outcome M4, M5 with adjustment of baseline value M1 (baseline model); for objective 2: regression model for difference score M4-M1, M5-M1, in statistic software (R [60], SPSS 22.0)); each model includes a categorical interaction between intervention group and pain class (i.e., CPG Pain Class dichotomized in grade 1 vs. 2–4). The models were controlled for age, sex, and study center. The significance of the interaction was calculated using the F-test for the interaction term. Additionally, for the presentation of a confidence interval, an equivalent regression model was applied to assess the effect between high pain patients in the SMT + BT group and high pain patients in the CG. In the statistic models, only complete cases for M1, M4, and M5 of the outcome variables were considered. Further, participants with more than 10 hours of physical activity/week (athletes) and those with a pain class of 0 were excluded.

The *p*-value refers to the interaction term and indicates whether the inclusion of the interaction term significantly improves the model adaptation, i.e., whether, as in this perspective, the value of the follow-up measurement for a high pain patient compared to a low pain patient depends on the intervention group. This means that a significant effect describes that the difference between high and low pain people differs statistically significantly between the intervention groups.

## 3. Results

### 3.1. Descriptive for Pain, Pain-Related Cognitions, and Mental Health

In total, *n* = 660 volunteers were randomly allocated in the randomized controlled trial. Of those, *n* = 439 participated in the two arms (SMT + BT and CG), of whom *n* = 291 (age 39.7 ± 12.7 years, female = 61.1%) were available as complete cases, a minimum pain class of 1, and hit all methodological requirements for M1, M4, and M5. Most participants suffered from a moderate pain intensity (CPI) and mild disability (DISS) of LBP (*n* = 223 with pain class = 1 and *n* = 68 with pain class >1). High standard deviations shown in Table 1 indicate that persons with higher pain were in the sample. Due to the necessity of performing physical activity, no severe (pain) cases were included.

Finally, 66% of all trial participants completed the study, indicating a good feasibility of this mixed, center-, and homebased program. There were no differences in baseline values between completers and noncompleters (n.s.).

### 3.2. Objective 1: Within Group Comparison

Table 2 and Figure 2 show the effect on the outcome for participants with higher pain compared to participants with lower pain within the given intervention group. Participants suffering from higher pain have on average 8.57 CPG points less pain intensity at M4 than participants with low pain within the multidisciplinary group. People with higher pain show improvements in mental health scales in comparison to lower pain patients, participating in the multidisciplinary group. This was true for symptoms of anxiety directly after the intervention and for anxiety and vital exhaustion at the half-year follow-up (M5). By contrast, people with higher pain compared to those with lower pain treated in regular routines have on average only a 2.34 CPG points improvement in pain intensity at M4.

### 3.3. Objective 2: Differences between the Multidisciplinary Intervention and the Control Group

The comparisons between difference scores of SMT + BT and CG are given in Figure 3 and Table 3. Regarding the profit of a multidisciplinary intervention for high pain persons in comparison to regular routines of the controls, it was shown that pain disability is more reduced, on average 11.81 CPG points (at the end of intervention) and on average 17.03 CPG points (at half-year follow-up) in the multidisciplinary group. Regarding mental health, people with higher pain profit more from the multidisciplinary intervention at both measurement points, while suffering less from symptoms of anxiety. The multidisciplinary group also seems to be more advantageous for other health complaints (exhaustion and inactivity). Again, the *p*-value refers to the interaction term, whereby only the patients with high pain are considered here.

Further information could be obtained by an additionally calculated equivalent regression model for anxiety (HADS anxiety M4: b = −3.54 (95%CI −5.56, −1.53), *p* = 0.001 and HADS anxiety M5: b= −2.88 (95%CI −5.21, −0.56), *p* =0.018) as well as for pain disability (DISS M4: b= −11.81 (95%CI −27.77, 4.16), *p* = 0.150 and DISS M5: b= −17.03 (95%CI −32.03, −2.04), *p* = 0.029). The coefficients, presented here in this text block, are identical to the group differences in Table 3.

## 4. Discussion

In objective 1, it was shown to what extent patients with more pain benefit from the different programs. For the synergetic multidisciplinary exercise approach, it was shown that people with higher pain profit from the reinforcing modules regarding mental health. This is true for a reduction of anxiety symptoms directly after the intervention as well as half a year later. Vital exhaustion can also be reduced within the first half-year and thus stabilizes basic physical recovery over the long term. Further, multidisciplinary therapy with mutually reinforcing mechanisms leads up to around 9 CPG points, showing a clinically relevant reduction of pain intensity for people suffering from higher pain [61,62,63,64]. By contrast, people with higher pain, with their regular routines (controls), only had a small reduction of pain intensity and disability and almost no changes in mental health complaints compared to those with lower pain. The *p*-value refers to the interaction term and indicates that a difference between high and low pain patients can not only be shown for the different variables between the intervention groups but also reach statistical significance for the reduction of anxiety and vital exhaustion in the multidisciplinary group.

In study objective 2, whether the self-management program is advantageous for patients with more pain in comparison to regular routines of controls was analyzed. It was shown that people with higher pain participating in the multidisciplinary group had a stronger reduction of pain disability and, in detail, around 12 CPG more at the end of the intervention and 17 CPG points at the half-year follow-up, same as those in regular routines in ambulatory settings. In addition, with regard to mental health concerns, the intervention group performed better in direct comparison with the control group across all areas (except stress), with a significantly stronger reduction in anxiety symptoms and sustainability.

In summary, it can be said that this new and innovative self-management exercise program for everyday life is a remarkable success. Of particular interest are the improvements in mental health complaints, which were intended but not directly expected in the context of a nonclinical-based form of training. Although improvements are small to moderate, they are all higher than in comparable studies with combined exercise and cognitive behavior therapy [65]. Positive effects on anxiety may be partially mediated by exercise itself [66] but are always an important player in limiting fear-avoidance to exercise, which is an underestimated problem in exercise programs [67]. The combination of reinforcing therapy components may therefore be the right approach to offer multidisciplinary therapy individually and yet broadly without long waiting times.

Regarding pain, in clinically relevant reductions between 12 and 17 CPG points and further, a good sustainability was shown in the homebased multidisciplinary program in direct comparison to regular routines [61,62,63,64]. Both the pain reduction and prolonged sustainability of the presented intervention narrow the earlier criticized gap of previous multidisciplinary programs [8,9,10]. The reported effects there were either not very consistent or of low sustainability.

The results raise hope that such a cost- and resource-saving program for low-back-pain people will be successful in both preventive and rehabilitative settings in the future. In respect of the health care system and the availability of multidisciplinary therapies only for patients in the advanced stages of LBP, this innovative home program can address a broad range of people at higher risk of chronification of pain after an intermittent low-back pain phase in primary and secondary prevention.

## 5. Conclusions

This study shows that a self-management program with reinforcing components could be of high clinical relevance in the treatment of unspecific low-back pain. Further, the presented program may be suitable for the medical supply in rural or socioeconomic weak areas. Generally, the capacity of exercise programs for therapeutic processes and their transfer to other medical supply settings such as rehabilitation aftercare is conceivable and raising hope for cost savings for the health system.

## 6. Limitations

First of all, the data presented in this study arose from a feasibility study which examined the practicability of conducting a homebound exercise intervention for the treatment of LBP. Starting problems finally led to the exclusion of one study center and a reduced sample size. The final, small sample size was lower than the power calculation of the pretest, strongly limiting significance levels. In respect of the clinically relevant improvements in pain and anxiety ratings and the debate around the clinical relevance of significance [68], the results still seem of high importance. Thirdly, for objective 2, the difference model was used—a standard routine in comparing groups (intervention vs. control). However, a difference model does not consider the entrance baseline attitude of pain; an overestimation of treatment effects may be possible. Lastly, we cannot rule out that people participating in self-management programs have higher motivation than others and that people with LBP are heterogeneous and vary in their chronicity, pain intensity, and activity level. It is discussed whether exercise treatments should be tailored to the individual capabilities and specific requirements of the patients, and this should be analyzed in future studies. The presented new treatment forms for LBP and the here-reported evidence seem of high potential for further improvements and for different prevention possibilities for LBP in different societies.

## Figures and Tables

**Figure 1 jcm-09-00115-f001:**
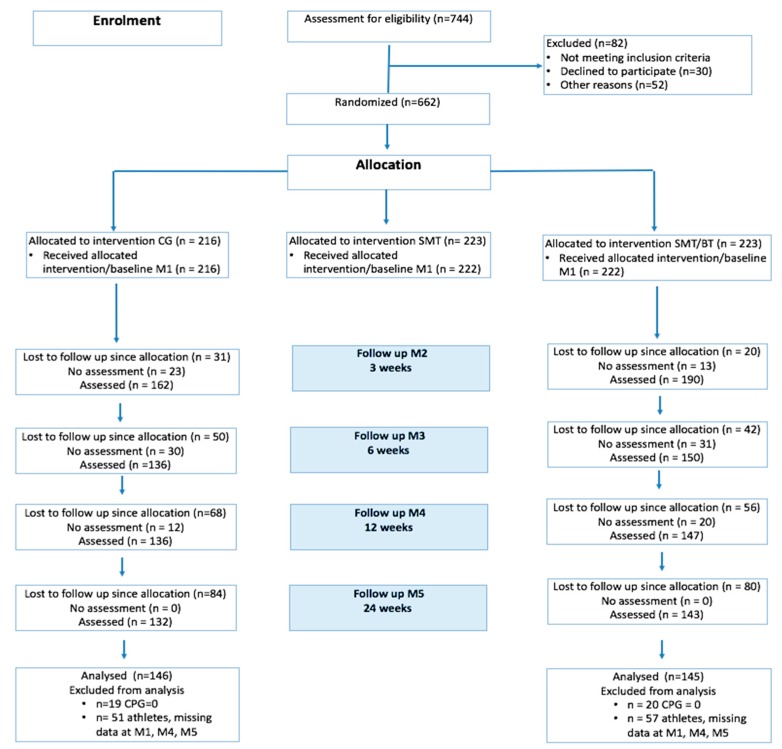
Study flow. (SMT: Sensorimotor Training, SMT/BT: Sensorimotor and Behavioral Training, CG: Control Group, CPG: Chronic Pain Grade).

**Figure 2 jcm-09-00115-f002:**
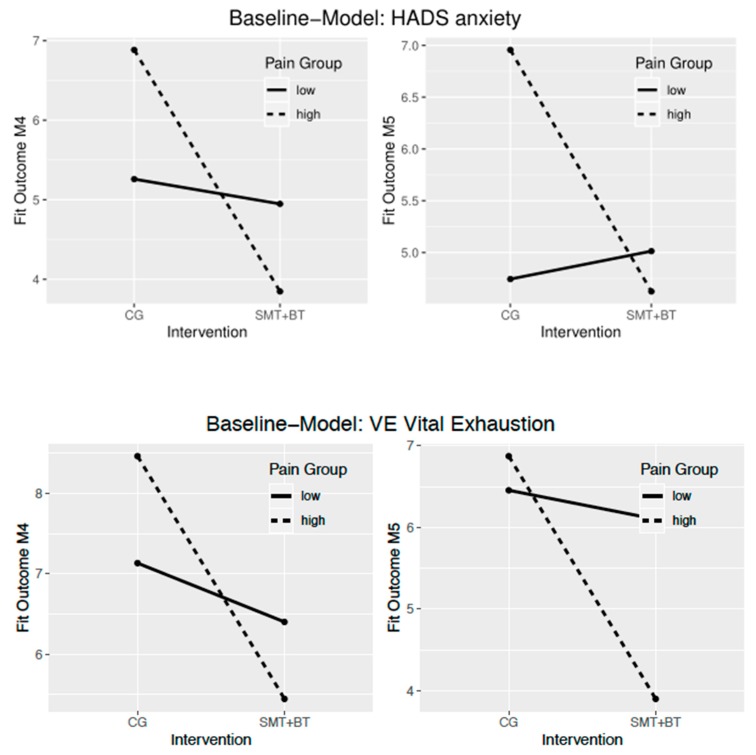
Baseline-Models for HADS anxiety (above) and vital exhaustion, showing an interaction effect of multidisciplinary treatment for high vs. low pain persons.

**Figure 3 jcm-09-00115-f003:**
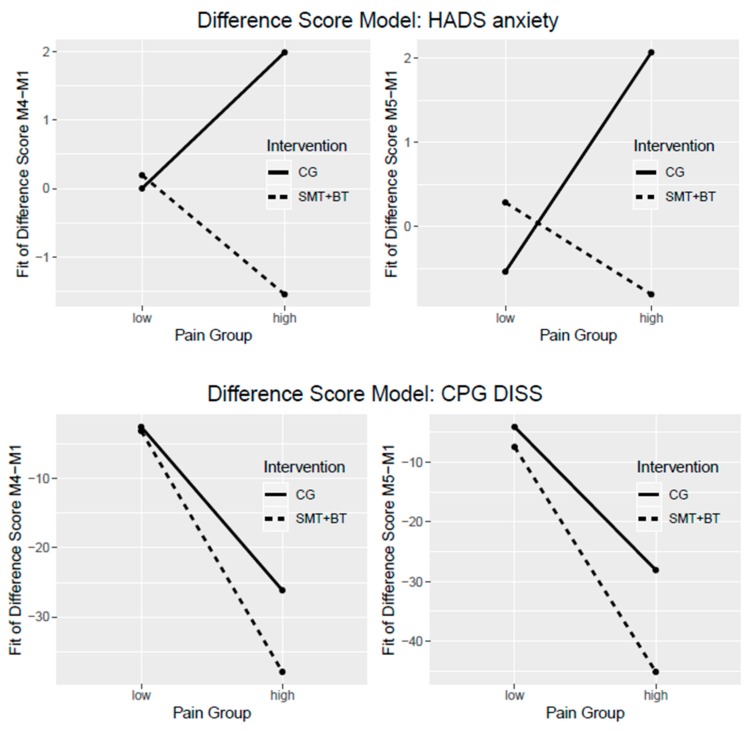
Difference score models for HADS anxiety (above) and subjective pain disability showing an interaction effect of treatments (multidisciplinary vs. regular routines) for persons with high pain.

**Table 1 jcm-09-00115-t001:** Descriptive characteristics (X **±** SD) at baseline (M1), 12 weeks (M4), and 24 weeks (M5) (*n* = 291).

Characteristic	%	X	SD	*n*	X	SD	*n*	X	SD	*n*
		M1			M4			M5		
Gender (female)	61.1	--	--	291						
Higher Education	40.53	--	--	227						
Living in partnership	55.95	--	--	227						
Age	--	39.7	12.7	291						
CPG CPI CG ^a^		32.81	19.35	146	28.43	20.07	83	25.42	17.69	83
CPG CPI SMT + BT		36.34	18.77	145	27.27	17.85	89	23.65	16.18	80
CPG DISS CG		17.05	21.93	146	10.32	17.11	83	8.05	12.65	83
CPG DISS SMT + BT		25.08	24.66	145	12.06	17.38	89	8.71	13.29	80
fab_activity CG ^b^		12.56	6.48	142	10.04	6.66	82	14.58	3.89	36
fab_activity SMT+ BT		12.86	5.29	139	13.01	5.85	88	12.26	5.71	53
HADS anxiety CG ^c^		5.17	3.04	144	5.99	3.20	80	5.80	3.25	79
HADS anxiety SMT + BT		5.23	3.04	142	4.63	2.86	82	4.63	3.02	70
HADS depression CG		3.69	2.99	144	3.77	3.33	80	3.79	3.06	77
HADS depression SMT + BT		3.81	2.97	143	3.13	3.08	84	3.18	3.30	71
PSS CG ^d^		16.09	6.47	139	16.55	5.65	80	14.25	5.59	79
PSS SMT + BT		15.88	5.77	137	14.59	5.93	86	13.57	7.37	76
VE CG ^e^		7.00	4.87	144	7.38	4.89	82	6.83	4.76	82
VE SMT + BT		7.57	5.15	141	6.15	4.72	85	6.18	5.08	79
PVAQ CG ^f^		37.27	13.02	146	34.18	13.55	83	--	--	--
PVAQ SMT + BT		38.10	12.53	145	34.15	11.68	89	--	--	--

^a^ CPG (Chronic Pain Grade scales): CPI characteristic pain intensity and DISS: subjective disability (score 0–100), ^b^ FABQ: pain-related cognitions scale: fab_activity (caused by bodily activity, score 0–30), ^c^ HADS-D: Anxiety and Depression (score 0–21), ^d^ PSS: Perceived Stress Scale (score 1–40), ^e^ vital exhaustion questionnaire (score 0–18), ^f^ PVAQ: Pain Vigilance and Avoidance Questionnaire (score 0–80); CG: Control Group, SMT + BT: multidisciplinary group. X: Mean Value, SD: Standard Deviation.

**Table 2 jcm-09-00115-t002:** Baseline–Models: differences in outcome between high and low pain participants (within group comparison).

Characteristics	M1 to M4			M1 to M5		
Groups, Test Statistic	CG	SMT + BT	F (*p*)	CG	SMT + BT	F (*p*)
CPG CPI ^a^	−2.34	−8.57	0.67 (0.415)	0.35	−2.46	0.15 (0.700)
CPG DISS	−3.40	−4.63	0.02 (0.878)	4.30	4.56	0.00 (0.964)
FABQ_activity ^b^	1.91	−1.23	0.91 (0.344)	0.59	−0.10	0.07 (0.786)
HADS anxiety ^c^	1.62	−1.10	5.96 (0.017) *	2.21	−0.39	4.00 (0.048) *
HADS depression	0.90	0.39	0.17 (0.683)	−0.36	0.02	0.09 (0.765)
PSS Perceived stress ^d^	−0.11	1.25	0.36 (0.552)	−0.64	0.98	0. 13 (0.579)
VE Vital Exhaustion ^e^	1.33	−0.95	1.70 (0.194)	0.42	−2.20	2.17 (0.144) ^#^
PVAQ Pain Vigilance Avoidance ^f^	0.47	1.41	0.06 (0.800)	--	--	--

*p* < 0.1 ^#^
*p* < 0.05 *; for the exact *p*-value for F-term, see in the brackets; the F-test shows significance of interaction terms. Multiple linear regression models with adjustment of baseline value, M1 to M4 and M1 to M5. Separate models for each scale, including categorical interaction between treatment group and pain class, controlled for age, gender, and study center. ^a^ CPG (Chronic Pain Grade scales): CPI characteristic pain intensity and DISS: subjective disability (score 0–100), ^b^ FABQ: pain-related cognitions scale: fab_activity (caused by bodily activity, score 0–30), ^c^ HADS-D: Anxiety and Depression (score 0–21), ^d^ PSS: Perceived Stress Scale (score 1–40), ^e^ vital exhaustion questionnaire (score 0–18), ^f^ Pain Vigilance and Avoidance Questionnaire (score 0–80).

**Table 3 jcm-09-00115-t003:** Difference score models between group differences in comparison to the control group. Presentation of the extent of profit for persons with higher pain in group comparison.

Characteristics	SMT + BT	F (*p*)	SMT + BT	F (*p*)
Differences in comparison to CG	M4–M1		M5–M1	
CPG CPI ^a^	−7.95	0.85 (0.359)	−6.24	0.34 (0.559)
CPG DISS	−11.81	1.55 (0.215)	−17.03	2.16 (0.109) ^#^
FABQ_activity ^b^	−0.11	0.72 (0.401)	−2.88	0.02 (0.892)
HADS anxiety ^c^	−3.54	10.55 (0.001) **	−2.88	7.70 (0.006) **
HADS depression	−1.44	0.46 (0.501)	−0.61	0.00 (0.993)
PSS Perceived stress ^d^	0.09	0.09 (0.768)	0.68	0.11 (0.746)
VE Vital Exhaustion ^e^	−3.06	1.37 (0.244)	−3.01	1.86 (0.195)
PVAQ Pain Vigilance Avoidance ^f^	−0.43	0.34 (0.560)	--	--

*p* < 0.1 ^#^, *p* < 0.01 **; for the exact *p*-value for the F-term, see in the brackets; the F-test shows significance of interaction terms. Multiple linear regression models: The models (separate models for each scale) with difference scores as outcome (M4–M1 and M5–M1) include categorical interaction between treatment group and pain class, controlled for age, gender, and study center. ^a^ CPG (Chronic Pain Grade scales): CPI characteristic pain intensity and DISS: subjective disability (score 0–100), ^b^ FABQ: pain-related cognitions scale: fab_activity (caused by bodily activity, score 0–30), ^c^ HADS-D: Anxiety and Depression (score 0–21), ^d^ PSS: Perceived Stress Scales (score 1–40), ^e^ Vital Exhaustion questionnaire (score 0–18), ^f^ Pain Vigilance and Avoidance Questionnaire (score 0–80).

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
