# Peer review of "The Feasibility and Effectiveness of a New Practical Multidisciplinary Treatment for Low-Back Pain: A Randomized Controlled Trial"

_jcm, 2019, doi:10.3390/jcm9010115_

Round 1

Reviewer 1 Report

Peer review – Journal of Clinical Medicine - jcm-674719

The feasibility and effectiveness of a new practical multidisciplinary treatment for low-back pain: a randomized controlled trial

Summary: This study evaluates the effects of an interdisciplinary sensorimotor exercise intervention compared to usual care in patients with low back pain on pain, pain related cognitions and mental health. The objectives were analyzed in 2 arms of a single-blind 3 armed randomized controlled trial (N744). The multidisciplinary intervention leads to a significantly stronger reduction of mental health complaints in people with higher pain than those with lower pain. Furthermore, people with higher pain participating in the multidisciplinary group show a significant bigger reduction of pain disability and mental health (anxiety) in comparison to controls

Major concerns: This is a large randomized controlled trial, studying a relevant intervention for a large group of patients for whom effective treatments are scarce. However, the study is incomplete in reporting and for that reason, the quality of the study is difficult to assess. I would advise the authors to follow the CONSORT Statement, a checklist of information to include when reporting a randomized controlled trial.

The abstract needs more details in the methods section: Please specify what treatments the control group get, instead of just reporting ‘control group’ Please specify the 9-week program in the abstract and the duration of follow-up Please specify clinical relevant outcome measures. What are clinically important differences? Briefly describe your population: mean age, gender, mean baseline pain score. Can you include the population size in Germany, to be able to place the 16-22 million euros in perspective? Do you have a reference for your statement in the second paragraph of the introduction that biopsychosocial factors should be taken into account as early as possible and that medication, operations and imaging should be handled prudently? Similar to the statement: “Secondly, the widely accepted biopsychosocial model of chronic pain not only suggests that biological, social and psychological factors contribute to the development of chronic pain, but also that there are interactions between these dimensions”. This requires a reference. The components of the interdisciplinary modules should be described in methods section instead of introduction. It’s described twice now. Please rephrase your objectives in PICO format (e.g. the objectives of the study are to evaluate the effects of an interdisciplinary sensorimotor exercise intervention compared to usual care in patients with low back pain on pain, pain related cognitions and mental health) Can you provide more information about the sample size calculation? Can you provide more information about the MiSpEx network, and the rational for the three armed RCT, and only using 2 arms for this study? The title suggests the reporting of the feasibility and the effectiveness of this multidisciplinary treatment. The outcomes measures are pain, pain related cognitions, and mental health, which relate to the effectiveness of the treatment. Can you separate the results related to the feasibility and effectiveness of the treatment? Who performed the randomization? What was the rationale for including patients with a minimum VAS score of 20 out of 100? Can you describe the variation in VAS scores between patients? Patients with a baseline VAS score or 20 are very different from patients with a baseline VAS score of 80 for example. Can you include a flow chart describing the numbers of participants who were randomly assigned, received intended treatment, and were analyzed for the primary outcome? Did you perform an intention to treat analysis? Can you include a table with baseline socio-demographic characteristics and outcomes at baseline? How many patients completed all follow-up, and how many patients didn’t? And is there a difference between the completers and non-completers in baseline characteristics and outcomes at baseline? Using linear regression models, the outcome should be the regression coefficient with a 95% confidence interval. Additionally, the number needed to treat and the unadjusted risk difference as absolute differences between groups should be calculated to provide easily interpretable effect estimated for medical specialists. Can you reflect on the generalizability of the results on the discussion? Is the population as well as the treatment generalizable to different countries and settings? A concluding paragraph is missing.

Author Response

Dear Reviewer,

We would like to thank you for having taken the time to read our manuscript. We are grateful for the opportunity to submit a letter and to revise manuscript parts for supplementary material. Attached you will find a detailed description of how we have addressed each comment specifically (see Þ as our answer)

Reviewer 1:

Summary: This study evaluates the effects of an interdisciplinary sensorimotor exercise intervention compared to usual care in patients with low back pain on pain, pain related cognitions and mental health. The objectives were analyzed in 2 arms of a single-blind 3 armed randomized controlled trial (N744). The multidisciplinary intervention leads to a significantly stronger reduction of mental health complaints in people with higher pain than those with lower pain. Furthermore, people with higher pain participating in the multidisciplinary group show a significant bigger reduction of pain disability and mental health (anxiety) in comparison to controls

Major concerns: This is a large randomized controlled trial, studying a relevant intervention for a large group of patients for whom effective treatments are scarce. However, the study is incomplete in reporting and for that reason, the quality of the study is difficult to assess. I would advise the authors to follow the CONSORT Statement, a checklist of information to include when reporting a randomized controlled trial.

Þ we already submitted a CONSORT Checklist, but obviously you did not received. We offer it again as supplementary material

Abstract

The abstract needs more details in the methods section: Please specify what treatments the control group get, instead of just reporting ‘control group’ Please specify the 9-week program in the abstract and the duration of follow-up Please specify clinical relevant outcome measures. What are clinically important differences? Briefly describe your population: mean age, gender, mean baseline pain score.

Þ we rewrote the abstract

Introduction

Can you include the population size in Germany, to be able to place the 16-22 million euros in perspective? Do you have a reference for your statement in the second paragraph of the introduction that biopsychosocial factors should be taken into account as early as possible and that medication, operations and imaging should be handled prudently? Similar to the statement: “Secondly, the widely accepted biopsychosocial model of chronic pain not only suggests that biological, social and psychological factors contribute to the development of chronic pain, but also that there are interactions between these dimensions”. This requires a reference.

Þ we integrated 83 millions people (line 47 ff)

Þ we integrated references at each place (line 55 ff)

The components of the interdisciplinary modules should be described in methods section instead of introduction. It’s described twice now.

Þ we re-structured the sections (can be seen in coloured first sentence lines within method part (header multidisciplinary intervention)

Please rephrase your objectives in PICO format (e.g. the objectives of the study are to evaluate the effects of an interdisciplinary sensorimotor exercise intervention compared to usual care in patients with low back pain on pain, pain related cognitions and mental health)

Þ We sort the objectives along PICOT (Person / Intervention / Comparison / Outcome of interest) (line 79ff)

Methods

Can you provide more information about the sample size calculation? Can you provide more information about the MiSpEx network, and the rational for the three armed RCT, and only using 2 arms for this study?

Þ Sample size calculation was based on a pilot study (unpublished material) in which the largest effect size for the primary outcome was selected. For this study effect size was conservatively completed to d=.20

Þ The MiSpEx network examined several thousand people over 8 years in three multicenter studies and 34 partial studies (see also references). We integrated a sentence for better explanation.

Mayer, F., et al., Medicine in Spine Exercise [MiSpEx] – A National Research Network to evaluate Back Pain in High-performance Sports as well as the General Population. German Journal of Sports Medicine, 2018. 69(7-8): p. 229-235.

Hönning, A., D. Stengel, and Güthoff, C., Statistical strategies to address main research questions of the MiSpEx network and meta-analytical approaches. German Journal of Sports Medicine, 2018. 69(7-8): p. 236-239

Þ firstly: multidisciplinary treatment is more appropriate for persons with higher or more complex pain syndromes and secondly: regular routines (can be also multidisciplinary, depending from the medical care giver’s strategy in ambulatory settings). To compare the same things, we focused only on a comparison within the multidisciplinary group (checking efficacy of reinforcing components) and between the multidisciplinary and control group (as regular routines can be multidisciplinary, but are mostly center based). For this reason we only present two arms here. The main effect and intention to treat analysis will be prepared in another paper, presenting all 3 arms.

Results

The title suggests the reporting of the feasibility and the effectiveness of this multidisciplinary treatment. The outcomes measures are pain, pain related cognitions, and mental health, which relate to the effectiveness of the treatment. Can you separate the results related to the feasibility and effectiveness of the treatment?

Þ we integrated this (line 243 ff)

Who performed the randomization?

Þ we used a randomisation list (nblock = 18, basis 1:1; www.randomization.com), see line 107 ff

What was the rationale for including patients with a minimum VAS score of 20 out of 100? Can you describe the variation in VAS scores between patients? Patients with a baseline VAS score or 20 are very different from patients with a baseline VAS score of 80 for example.

Þ VAS was used as a quick inclusion tool, to be sure, that people which were included, suffering from pain. Primary outcome was afterwards measured by questionnaires (an here with the chronic pain grade questionnaire representing disability due to pain and characteristic pain intensity in the last three months). Although VAS is often used (also in Cochranes), it is only a momentary picture, very instable (also influenced by daily mood states) and not appropriate to use it as measure for primary outcomes of intervention efficacy.

Þ To give more information about the persons under study we offer now the numbers of people with a pain grade > 0 in a study flow chart.

Can you include a flow chart describing the numbers of participants who were randomly assigned, received intended treatment, and were analyzed for the primary outcome?

Þ we integrated a study flow chart (line 120)

Did you perform an intention to treat analysis?

Þ please see above. This is already planned by network members, also offering whole information about main effects between the three groups.

Can you include a table with baseline socio-demographic characteristics and outcomes at baseline?

Þ This is already included. Please see table 1

How many patients completed all follow-up, and how many patients didn’t? And is there a difference between the completers and non-completers in baseline characteristics and outcomes at baseline?

Þ 66% completed all measurements (we integrated this information see 246f). But this does not mean, that we have 66% persons with complete cases.

Using linear regression models, the outcome should be the regression coefficient with a 95% confidence interval. Additionally, the number needed to treat and the unadjusted risk difference as absolute differences between groups should be calculated to provide easily interpretable effect estimated for medical specialists.

Þ Each regression model contains an interaction between two binary variables (low vs. high pain group; CG vs. SMT+BT), resulting in three regression coefficients, P-values and confidence intervals, which independently are not interpretable. Therefore we choose to present the differences of the fitted outcome values of each group contained in the interaction of the regression model, as well as the significance of the interaction, in the tables and exemplary figures

Can you reflect on the generalizability of the results on the discussion? Is the population as well as the treatment generalizable to different countries and settings?

Þ.we integrated a small section in the conclusion part (line 328)

A concluding paragraph is missing.

Þ we integrated a concluding paragraph

Reviewer 2 Report

Review of “ The feasibility and effectiveness of a new practical multidisciplinary treatment for low-back pain: a randomized controlled trial”

The stated aim of the present study is timely and appropriate given the global impact of LBP which is now recognized as the number one musculoskeletal condition impacting on quality of life and resulting in severe socioeconomic impact not only on the afflicted individuals but also severely effecting the welfare of their families. Effective methodologies certainly need to be developed to address this most debilitating of conditions and provide measures to counter its symptoms which severely impact globally on healthcare resources in all countries.

When I looked at the data presented it was difficult to see what the authors had actually done and what the results really meant.   The data is not presented in a manner that allows the significance of the results to be readily ascertained. The tables have so many acronyms that they are almost impossible to understand. Confidence intervals and ranges are important data to present and these apparently are not covered. It appears some of the data presented is scores which may or may not be normally distributed or even be continuous data. For these, medians and ranges are more appropriate data to present and logistic regression rather than linear regression is a more usual treatment for such data. For so much data a lack of graphical presentations is unusual. I suspect this is because the data is so variable and this brings into question in my mind how useful the scheme is in practical terms. I would have liked to see graphical analyses of the most significant findings.

I would recommend that the data be reorganised in such a manner as to convince this reviewer of the utility of their scheme. As presented the data is rather confusing and is not user friendly. Maybe it would be more appropriate that this manuscript be submitted to a physiotherapy journal.

Minor points

Line 36/37 “ being physical active may benefit of homebased low-cost exercise programs,” English should be improved maybe as being physically active may benefit home-based low-cost exercise programs.

Line 46/47 The statement “Disabilities due to LBP are expected to increase most in middle- income countries due to lifestyle changes such as sedentary occupations or obesity trends, and limited access to quality health care” may need re-wording. Surely, low income countries would be most adversely affected in terms of treatment availability. These countries are also impacted by adverse ageing and obesity demographics according to the World Bank from data collected by the United Nations from 28 countries. Global reviews of LBP show this condition has socio-economic impact across all countries irrespective of their economic status, this is not just a disease of more economically advanced countries. However this reviewer can see why the stated sample group is primarily of interest in the present study since this is most appropriate for German population groups.

Line 70 onwards – a change in font size is noted. The whole manuscript should have a uniform font type and size throughout the textstream.

Line 77, “and evidence about a specific form of exercise superior to another is lacking”- English needs improvement.

Line 152   (for an overview of the overall study design. see (38)). This information or a brief summary of it should have been provided in the current manuscript.

Line 233 define RCT

Line 234 define ‘f’

Line 313-314 “presented data originate from a feasibility study which primarily aimed at the evaluation of the practicability of a homebased exercise intervention.”. English improvement required, maybe the data presented herein arose from a feasibility study which examined the practicality of conducting a home-bound exercise intervention for the treatment of LBP.

Statistical format. The correct format for significance P values is a CAP Italic not small p non italic. The number of samples examined is normally given as a small ‘n’, ie n=12 like used in lines 236-238. These terms should be used in a consistent manner.

Statistical data In linear regression it is conventional to provide this data as mean (X) ± Standard deviation (SD) or standard error of the mean (SEM) with the number of samples (n) examined supplied for each group and the significance between two sample groups being compared indicated with a P value, the actual P value should be specified and not just P<0.05.

Table 1. Define M and SD are these mean and standard deviation?   What was the statistical significance between comparisons of the M1, M4 and M5 groups of data for each parameter and between parameters within each sample group?

Table 3 CG and CGI should be defined at first point of use. Define SMB + BT and F. Clarify HADS-D definition to explain HADS anxiety and HADS depression terms

Tables 1-3 . Headings above data columns should be aligned above the data they refer to.

Author Response

Dear Reviewer,

We would like to thank you for having taken the time to read our manuscript. We are grateful for the opportunity to submit a letter and to revise manuscript parts for supplementary material. Attached you will find a detailed description of how we have addressed each comment specifically (see Þ as our answer)

Reviewer 2

The stated aim of the present study is timely and appropriate given the global impact of LBP which is now recognized as the number one musculoskeletal condition impacting on quality of life and resulting in severe socioeconomic impact not only on the afflicted individuals but also severely effecting the welfare of their families. Effective methodologies certainly need to be developed to address this most debilitating of conditions and provide measures to counter its symptoms which severely impact globally on healthcare resources in all countries.

When I looked at the data presented it was difficult to see what the authors had actually done and what the results really meant. The data is not presented in a manner that allows the significance of the results to be readily ascertained. The tables have so many acronyms that they are almost impossible to understand. Confidence intervals and ranges are important data to present and these apparently are not covered. It appears some of the data presented is scores which may or may not be normally distributed or even be continuous data. For these, medians and ranges are more appropriate data to present and logistic regression rather than linear regression is a more usual treatment for such data. For so much data a lack of graphical presentations is unusual. I suspect this is because the data is so variable and this brings into question in my mind how useful the scheme is in practical terms. I would have liked to see graphical analyses of the most significant findings.

I would recommend that the data be reorganised in such a manner as to convince this reviewer of the utility of their scheme. As presented the data is rather confusing and is not user friendly. Maybe it would be more appropriate that this manuscript be submitted to a physiotherapy journal.

Þ We re-arranged the data presentation

Acronyms in the table 2, 3 and text were clarified according to your comments about statistical format and data presentation (replace of p P / n n / M(SD) mean (X) ± Standard deviation (SD) We calculated additionally confidence intervals for the regression coefficients of the effect between high pain patients in SMT+BT and high pain patients in CG. integrated Confidence intervals (CI) and Cohen’s d for a better estimation of the clinical relevance. Presented interval scale data are continuously outcome variables. Therefore we choose linear regression and not logistic regression for binary outcome variables. As suggested we added figures for the significant results (see line 258, 277ff). As the persons under study suffer from low to moderate and not serious pain, the variability is higher as seen in SD. But results giving notice of a high impact for early treatment or prevention before developing comorbidity states with seriously pain problems.

Minor Points

Line 36/37 “ being physical active may benefit of homebased low-cost exercise programs,” English should be improved maybe as being physically active may benefit home-based low-cost exercise programs.

Þ revised

Line 46/47 The statement “Disabilities due to LBP are expected to increase most in middle- income countries due to lifestyle changes such as sedentary occupations or obesity trends, and limited access to quality health care” may need re-wording. Surely, low income countries would be most adversely affected in terms of treatment availability. These countries are also impacted by adverse ageing and obesity demographics according to the World Bank from data collected by the United Nations from 28 countries. Global reviews of LBP show this condition has socio-economic impact across all countries irrespective of their economic status, this is not just a disease of more economically advanced countries. However this reviewer can see why the stated sample group is primarily of interest in the present study since this is most appropriate for German population groups.

Þ we revised the sentence see line 45-49.

Line 70 onwards – a change in font size is noted. The whole manuscript should have a uniform font type and size throughout the textstream.

Þ we revised it and proved font size for the whole manuscript.

Line 77, “and evidence about a specific form of exercise superior to another is lacking”- English needs improvement.

Þ we will request the English editing service from MDPI if the manuscript will be accepted

Line 152   (for an overview of the overall study design. see (38)). This information or a brief summary of it should have been provided in the current manuscript.

Þ we integrated a sentence and further, study flow (line 120)

Line 233 define RCT

Þ we revised it (line 234).

Line 234 define ‘f’

Þ we revised it (line 235)

Line 313-314 “presented data originate from a feasibility study which primarily aimed at the evaluation of the practicability of a homebased exercise intervention.”. English improvement required, maybe the data presented herein arose from a feasibility study which examined the practicality of conducting a home-bound exercise intervention for the treatment of LBP.

Þ we revised it and will give it to a MDPI proof reading, if the manuscript will be accepted

Statistical format. The correct format for significance P values is a CAP Italic not small p non italic. The number of samples examined is normally given as a small ‘n’, ie n=12 like used in lines 236-238. These terms should be used in a consistent manner.

Þ we revised the statistical terms in tables and text.

Statistical data In linear regression it is conventional to provide this data as mean (X) ± Standard deviation (SD) or standard error of the mean (SEM) with the number of samples (n) examined supplied for each group and the significance between two sample groups being compared indicated with a P value, the actual P value should be specified and not just P<0.05.

Þ we revised the statistical terms in tables and text. Exact P-value was shown in table 2 and 3. For further clarifying, we noted it also at the bottom of each table.

Þ Each regression model contains an interaction between two binary variables (low vs. high pain group; CG vs. SMT+BT), resulting in three regression coefficients, p-values and confidence intervals, which independently are not interpretable. Therefore we choose to present the differences of the fitted outcome values of each group contained in the interaction of the regression model, as well as the significance of the interaction, in the tables and exemplary figures.

Table 1. Define M and SD are these mean and standard deviation?   What was the statistical significance between comparisons of the M1, M4 and M5 groups of data for each parameter and between parameters within each sample group?

Þ M (mean) and SD (standard deviation are now presented as X and SD

Þ intention to treat analysis (group differences and longitudinal effects of the intervention on outcome parameters) will be presented in an own article, which is planned of network members.

Table 3 CG and CGI should be defined at first point of use. Define SMB + BT and F. Clarify HADS-D definition to explain HADS anxiety and HADS depression terms

Þ CG, SMT+BT as well as SMT were introduce in design part now, see line140 ff

Þ HADS and F were already described in instrument and statistical analysis part. But for better readability we now also mentioned it additionally at the bottom of each table.

Tables 1-3 . Headings above data columns should be aligned above the data they refer to.

Þ we revised it

Round 2

Reviewer 1 Report

The authors have sufficiently revised the manuscript, and processed the comments from the first revision. I only have 3 more outstanding questions:

Consort requires the reasons for exclusions in the flowchart. Can you include those for each point in time? It is still unclear if you have performed an Intention To Treat analysis. Can you include socio-demographic characteristics in Table 1? Information about age, gender, etc will give the reader a better idea of the included population.

Author Response

Revision 2

Minor changes:

Consort requires the reasons for exclusions in the flowchart. Can you include those for each point in time?

We integrated a new flow chart

It is still unclear if you have performed an Intention To Treat analysis.

We did not an Intention to Treat Analysis, because we focused on people with pain (only people with CPG >0 were included in our analysis). An intention to treat analysis would include all people (also those with pain class 0). This is mentioned in the method part, see line 228-230

Can you include socio-demographic characteristics in Table 1? Information about age, gender, etc will give the reader a better idea of the included population.

We integrated age, gender, education and family status in the table

Minor English spell check

A native speaker did a proof read

Reviewer 2 Report

Review of   the revised manuscript The feasibility and effectiveness of a new practical multidisciplinary treatment for low-back pain: a randomized controlled trial.

The revised manuscript is considerably improved and reads well. The graphical figures are especially welcome additions and clearly show the effectiveness of the proposed interventions thus the impact of the revised manuscript is readily discernable to readers. The authors have addressed all of my suggestions and the manuscript is a welcome and worthy addition to the literature.

Author Response

Thank you very much for your time and supporting comments!